# Genome editing for scalable production of alloantigen-free lentiviral vectors for *in vivo* gene therapy

Michela Milani[1,2], Andrea Annoni[1], Sara Bartolaccini[1], Mauro Biffi[1], Fabio Russo[1], Tiziano Di Tomaso[1], Andrea Raimondi[3], Johannes Lengler[4], Michael C Holmes[5], Friedrich Scheiflinger[4], Angelo Lombardo[1,2], Alessio Cantore[1,†] & Luigi Naldini[1,2,*,†] (iD)

## Abstract

Lentiviral vectors (LV) are powerful and versatile vehicles for gene therapy. However, their complex biological composition challenges large-scale manufacturing and raises concerns for *in vivo* applications, because particle components and contaminants may trigger immune responses. Here, we show that producer cell-derived polymorphic class-I major histocompatibility complexes (MHC-I) are incorporated into the LV surface and trigger allogeneic T-cell responses. By disrupting the beta-2 microglobulin gene in producer cells, we obtained MHC-free LV with substantially reduced immunogenicity. We introduce this targeted editing into a novel stable LV packaging cell line, carrying single-copy inducible vector components, which can be reproducibly converted into high-yield LV producers upon site-specific integration of the LV genome of interest. These LV efficiently transfer genes into relevant targets and are more resistant to complement-mediated inactivation, because of reduced content of the vesicular stomatitis virus envelope glycoprotein G compared to vectors produced by transient transfection. Altogether, these advances support scalable manufacturing of alloantigen-free LV with higher purity and increased complement resistance that are better suited for *in vivo* gene therapy.

**Keywords** gene therapy; hemophilia; lentiviral vectors; MHC-I; stable producer cell line

**Subject Categories** Genetics, Gene Therapy & Genetic Disease

## Introduction

Gene therapy has recently shown remarkable progress in clinical trials and promises to provide effective treatment for several genetic and acquired diseases (Naldini, 2015). Underlying this success is the development of improved gene transfer vectors (Kay, 2011; Mingozzi & High, 2011; Nathwani *et al*, 2014). Among them, lentiviral vectors (LV) are emerging as versatile vehicles of relatively large capacity for stable transgene integration in the genome of target cells. LV are currently exploited both for *ex vivo* gene therapy, in which target cells (such as hematopoietic stem/progenitors cells, HSPC or T cells) are harvested from the patient, transduced, and then re-infused, and for *in vivo* gene therapy, in which LV are directly injected into the patient, either into the bloodstream or *in situ*, such as in the brain (Palfi *et al*, 2014). Whereas several ongoing clinical trials support the efficacy and safety of LV for *ex vivo* gene therapy (Cartier *et al*, 2009; Cavazzana-Calvo *et al*, 2010; Aiuti *et al*, 2013; Biffi *et al*, 2013; Maude *et al*, 2014), *in vivo* liver-directed gene therapy with LV remains more challenging. Indeed, LV particles undergo a complex assembly with the outer envelope deriving from the membrane of packaging cells, thus comprising an array of proteins beside the viral antigens that may act as immune triggers upon recognition and phagocytosis by professional antigen presenting cells (APC; Annoni *et al*, 2013). Moreover, current manufacturing of LV mostly relies on transient plasmid co-transfection, which is labor-intensive and poorly amenable to scale-up and standardization, and leads to substantial amount of impurities, including residual plasmid DNA, which may further aggravate APC activation. In addition, the reported instability of vesicular stomatitis virus glycoprotein G (VSV.G)-pseudotyped LV in human serum remains a concern for *in vivo* administration (DePolo *et al*, 2000; Croyle *et al*, 2004; Trobridge *et al*, 2010; Hwang & Schaffer, 2013). We have previously reported that, upon

1 San Raffaele Telethon Institute for Gene Therapy, IRCCS San Raffaele Scientific Institute, Milan, Italy
2 Vita Salute San Raffaele University, Milan, Italy
3 IRCCS San Raffaele Scientific Institute, Milan, Italy
4 Baxalta (former Baxter) Innovation GmbH, Vienna, Austria
5 Sangamo Therapeutics, Inc., Richmond, CA, USA
*Corresponding author. Tel: +39 02 2643 4681; E-mail: luigi.naldini@hsr.it
†These authors share senior authorship

intravenous administration, LV allow stable gene transfer to the liver, provided that transgene expression is stringently targeted to hepatocytes, and have shown dose-dependent therapeutic efficacy in a mouse and a canine model of hemophilia B, a coagulation disorder due to mutations in the factor IX (FIX)-encoding gene (Brown *et al*, 2006; Matrai *et al*, 2011; Cantore *et al*, 2012, 2015). Despite the therapeutic potential, some hurdles remain to be addressed before considering clinical translation of *in vivo* LV administration, such as the manufacturing of sufficiently large, consistent, and highly purified batches for *in vivo* delivery, the vector stability in the circulation, and the risk of acute toxicity and immunogenicity triggered by particle components or contaminants. Here, we describe an inducible scalable packaging cell line, which supports consistent generation of high-yield producers of LV of interest by a targeted integration strategy. LV produced by these cells achieve equivalent levels of gene transfer in the liver and are stable upon concentration and purification as LV produced by conventional transfection, but are more resistant to inactivation in human sera and lack plasmid DNA contaminants. Moreover, by further editing the genome of LV producer cells, we modified the protein composition of their plasma membrane and in turn of the LV envelope and obtained novel LV with enhanced capacity to escape immune recognition, which are better suited for *in vivo* applications.

## Results

### Reproducible generation of LV producer cell lines by targeted integration

In order to avoid toxicity due to stable expression of viral components, we took advantage of a regulated, tetracycline (Tet)-dependent system, in which a Tet-regulated transcriptional repressor (Tet-R) binds to DNA sequences included in a promoter and represses transcription by steric hindrance (Yao *et al*, 1998; Jones *et al*, 2005). Upon addition of doxycycline (dox), Tet-R is released, allowing transcription. To generate LV producer cell lines, we thus started from a 293 cell line stably expressing Tet-R. We then step-wise transfected the plasmids expressing third-generation LV components, human immunodeficiency virus type 1 (HIV) Rev, Gag/Pol, and the VSV.G pseudotype (Dull *et al*, 1998), under the control of Tet-regulated promoters and coupled with antibiotic resistance cassettes (Fig 1A and B). We first introduced Rev, showed inducible Rev expression in 32/33 clones analyzed (Appendix Fig S1A), then pooled > 4,000 clones, and stably transfected Gag/Pol. We then screened clones for inducible LV particle production (Appendix Fig S1B), pooled the six highest expressing clones, and stably transfected VSV.G. We confirmed inducible Rev and VSV.G expression (Appendix Fig S1C) and pooled the highest expressing clones to obtain the packaging cell line. This line yielded $66 \pm 14$ ng of HIV Gag p24/ml after dox addition, with > 500-fold induction over a very low basal level (Fig 1C and Appendix Fig S1B), a productivity that, although in the lower range of that obtained by transient transfection, prompted us to continue development of this cell line. We found one copy of *Rev, Gag,* and *VSV.G* DNA per genome in the packaging cell line (Fig 1D), suggesting that integration site selection rather than copy accumulation played a role in the higher expression. We thus adopted site-specific integration as an efficient and reproducible means to introduce a full-length, self-inactivating (SIN)-LV genome transfer construct (Zufferey *et al*, 1998; Follenzi *et al*, 2000), and convert the packaging into a producer cell line (see Fig 1B). We targeted the adeno-associated virus site 1 (*AAVS1*), previously described as permissive to high-level stable expression and tolerant to integration, exploiting site-specific endonucleases and homology-directed repair (Lombardo *et al*, 2011). We included a GFP selector preceded by a splice acceptor site and a sequence encoding the self-cleaving 2A peptide after the LV 3′ long terminal repeat (Figs 1E and EV1A). Because integration occurs in the first intron of the *PPP1R12C* gene,

**Figure 1. Generation and molecular characterization of LV producer cell lines.**

A   Schematic representation of the plasmids expressing third-generation LV packaging components (HIV Rev, Gag/Pol) and the surface glycoprotein of the vesicular stomatitis virus, VSV.G (pseudotype; Dull *et al*, 1998), coupled with antibiotic resistance cassettes, used to generate the LV packaging cell line. CMV-2xTetO₂, immediate/early enhancer/promoter of cytomegalovirus (CMV) with two tetracycline operator elements (TetO₂); BGH pA, bovine growth hormone polyadenylation signal; SV40, simian virus 40 promoter; SV40 pA, simian virus 40 polyadenylation signal; SD, splice donor site; SA, splice acceptor site.

B   Flowchart of the generation of LV packaging and producer cell lines. Rev, Gag/Pol, and VSV.G-expressing plasmids were introduced into a 293 cell line stably expressing a tetracycline-regulated transcriptional repressor (293 T-REx; Yao *et al*, 1998) by subsequent rounds of transfection and antibiotic selection, to obtain the packaging cell line. Further genome engineering allows modifying the packaging cell line for the desired features. Targeted integration of a LV genome transfer construct allows consistent generation of producer cell lines of LV of interest. GOI, gene of interest.

C   LV physical particle content (ng of HIV Gag p24/ml) in medium collected from the packaging cell line 3 days after dox induction.

D   DNA copies of *Reu* (pink bar), *Gag* (gray bar) or *VSV.G* (blue bar) per diploid genome in the packaging cell line.

E   Schematic representation of the plasmid used as donor DNA (pLV) for homologous recombination (top) to target the LV genome transfer construct into *AAVS1* (bottom), which is found within the first intron of the *PPP1R12C* gene (see also Fig EV1A). Brown and light blue arrows represent the sequences homologous to the genomic target site. The HIV U3 region of the 5′ long terminal repeat (LTR) is replaced by the CMV promoter/enhancer allowing synthesis of the full-length RNA for packaging (Dull *et al*, 1998). The HIV enhancer/promoter was deleted from the 3′ LTR (ΔU3), thus obtaining SIN LV (Zufferey *et al*, 1998). Ψ, packaging signal; Prom, internal promoter; wpre, woodchuck hepatitis virus post-transcriptional regulatory element (Zufferey *et al*, 1999; Zanta-Boussif *et al*, 2009); 2A, porcine teschovirus-1 2A sequence. The black arrow shows transcription of the locus, and the brown and light blue arrows represent the primers used to detect the LV genome junctions.

F–H   PCR analyses (F) for the 5′ and 3′ LV genome junctions generated by targeted integration (T.I.) of the donor DNA into the locus, or DNA copies of LV genome construct per diploid genome (G, H) in bulk GFP-positive (+) or GFP-negative (−) sorted populations and single-cell clones obtained from three independent T.I. experiments performed with the indicated donor DNA (see also Fig EV1A). (F) Red borders show images taken from different gels.

Data information: In (C), data are presented as mean with standard error of the mean, SEM, of three independent inductions.
Source data are available online for this figure.

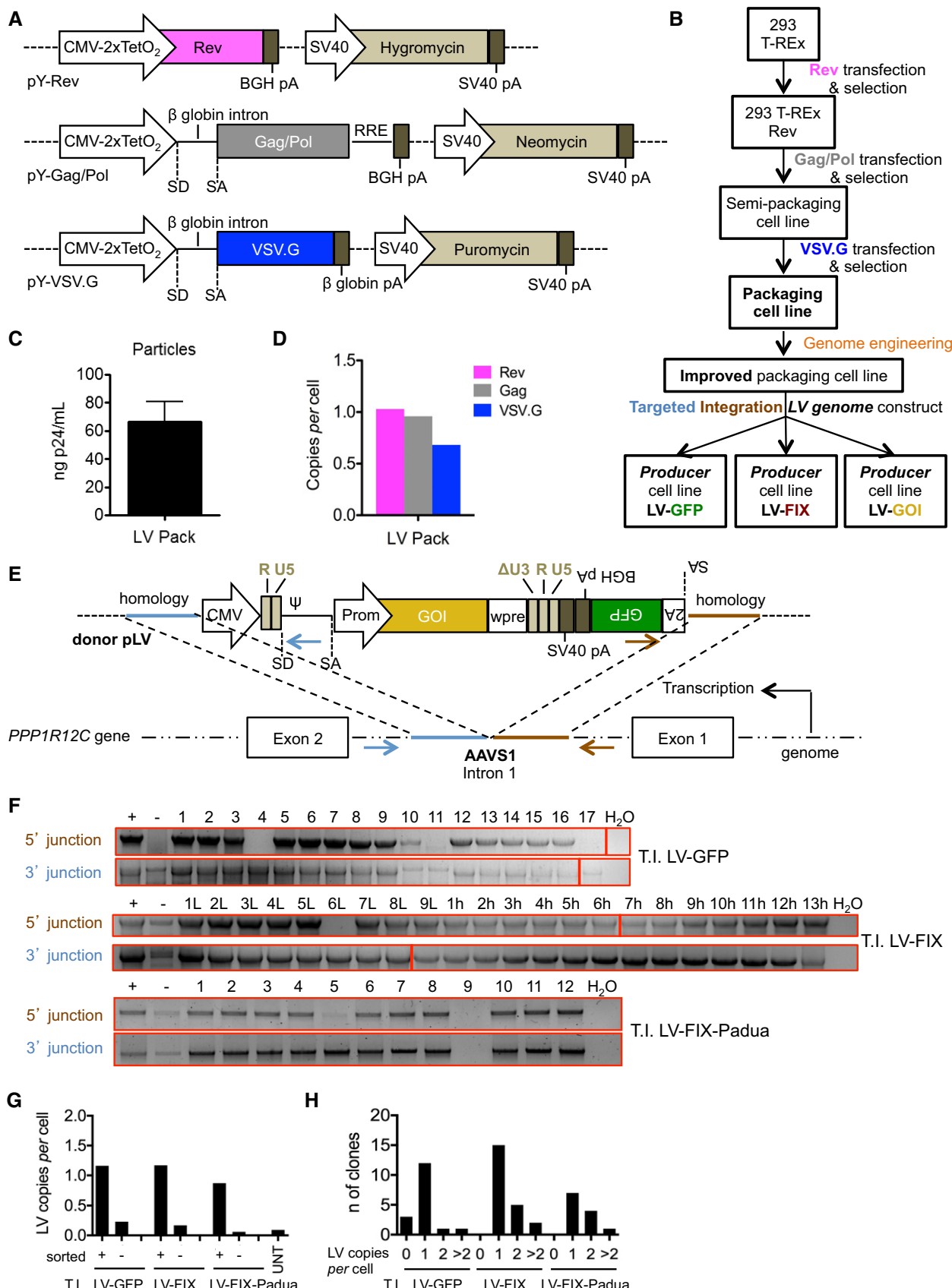

**Figure 1.**

GFP expression originates from the endogenous promoter (Lombardo *et al*, 2011), allowing selection of targeted cells. We performed three independent targeted integrations (T.I.) of LV constructs expressing GFP or human FIX, either wild-type (wt) or a codon-optimized hyper-functional version (FIX-Padua; Fig EV1A; Cantore *et al*, 2015), by transiently co-delivering zinc-finger nucleases (ZFN) targeting *AAVS1* and the plasmid donor DNA. We achieved between 2 and 5% of GFP-positive cells, then enriched the GFP-positive cells by fluorescence-activated cell sorting (FACS), and obtained bulk and several single-cell-derived clones (*n* = 51) for each targeting (Fig EV1B). All clones analyzed (43/43 that grew well in culture among the 51 clones) were GFP positive, with some variation in mean fluorescence intensity (MFI) which was lower in those targeting in which GFP expression relied on splicing and 2A-mediated protein release, as expected (Fig EV1C). All clones except for one showed one copy of *Rev, Gag,* and *VSV.G* DNA per genome and no integration of ZFN DNA (Fig EV1D and E); the majority of the clones (44/51) presented the two expected *AAVS1*-LV genome junctions by PCR (Fig 1F), 34 clones (67%) had one LV copy per cell, 10 (20%) had two, four (8%) > 2, and three (6%) had none (Fig 1G and H). Together, these data show a high rate of mono-allelic on-target integration of this strategy, as previously reported for different purposes (Lombardo *et al*, 2011).

### Robust and scalable LV production by stable producer cell lines

We measured infectious titer, physical particles, and specific infectivity of LV-containing supernatant of dox-induced bulk-sorted positive populations and single-cell clones, selected for robust growth rate, for the three T.I. experiments. At peak production, 3 days after induction (Fig EV2A and B), we found on average $1.5 \times 10^6$ transducing units (TU)/ml, 56 ng/ml p24 equivalents, $3 \times 10^4$ TU/ng p24, reaching up to $4.4 \times 10^6$ TU/ml, 222 ng/ml p24/ml, and $8 \times 10^4$ TU/ng p24 (Fig 2A). The infectivity of cell line-produced LV was in the lower-bound range of that obtained for LV produced by transient transfection in our standard conditions. We compared the LV output of several different inductions of the bulk-sorted LV-GFP producer cells over the course of 1 year and observed that productivity is maintained also after a freeze/thaw cycle and even without antibiotic selective pressure (Fig 2B). Lentiviral vectors production was scaled in cell factories up to 6 liters with similar or higher output in the collected conditioned medium than measured in small-scale experiments (Table 1), and this medium was processed by our previously reported two-step chromatography purification (Biffi *et al*, 2013), giving the expected final yield of vector. Specific infectivity was maintained both throughout the purification process and after concentration by ultracentrifugation (Fig EV2C). Concentrated LV was stable at −80°C after 10 months of storage (Fig EV2D). Overall, these data indicate comparable stability of LV particles produced by the cell line or by transient transfection. We found undetectable titer (< 100 TU/ml) and 120 pg p24/ml in medium collected from the LV-GFP producer cell line in the absence of dox (see also Appendix Fig S1B), suggesting that the low amount of LV particles produced are not infectious, possibly because the low levels of Rev in the non-induced condition limit nuclear exports of full-length LV genome for encapsidation. These data are consistent with the reported low leakiness of the Tet-R system and support the long-term stability observed for the cell line, which may be protected from the toxicity of viral proteins and from LV superinfection in the non-induced state. Overall, these data show that single-copy integration into *AAVS1* mediates robust transcription of the LV genome and the generation of highly infectious vector particles.

We then transduced human cord blood-derived HSPC with concentrated LV produced by the two most productive clones of the LV-GFP producer cell line, or by transient transfection. We observed a LV dose-dependent increase in the percentage of transduced cells and vector copies per diploid genome (vector copy number, VCN), reaching up to 45% GFP-positive and 1.9 VCN in the cultured cells, 70% GFP-positive cells and 2.8 VCN in CFC assay, at the highest multiplicity of infection (MOI) of the cell line-produced LV, and a two- to fivefold lower dose–response than observed for transient transfection LV (Fig 2C–F). The percentage of erythroid (CD235a-positive) and myeloid (CD33-positive) cells among the total CFC did not change significantly among the different transductions (Fig EV3A–C). The lower transduction efficiency of cell line-derived than transient transfection-derived LV at matched MOI likely reflects the lower specific infectivity of the former vector. However, the cell line-produced LV still allowed reaching clinically relevant VCN in the HSPC (Aiuti *et al*, 2013; Biffi *et al*, 2013). We also transduced activated primary human T cells and achieved > 90% transduction and VCN > 5 at the highest MOI of LVs produced by either method (Fig 2G and H). We did not observe any skewing in the CD4/CD8

---

**Figure 2. Evaluation of LV produced by the producer cell lines.**

A, B LV infectious titer (TU/ml, black bars or line, plotted on left *y*-axis), physical particles (ng p24/ml, dashed bars or line, plotted on right *y*-axis), and specific infectivity (TU/ng p24, gray bars or line, plotted on left *y*-axis) in conditioned medium of (A) bulk GFP-positive (+) sorted populations and single-cell clones obtained from three independent T.I. experiments performed with the indicated donor DNA, 3 days after dox induction, or (B) at the indicated time (months) of continuous culture in the absence of antibiotic selective pressure. Axis interruption indicates a freeze/thaw cycle. The *n* of independent inductions of LV production from bulk-sorted populations (+) or single-cell clones is shown on top of the bars in panel (A), when not 1.

C–F Percentage of GFP-positive cells (C, D) and VCN (E, F) in the CD34-positive cells culture (C, E) or pooled colonies (D, F) from CFC assays (MOI 10 and 100, *n* = 4 transductions in 2 independent experiments using three different LV batches per production method; MOI 300, *n* = 2 transductions). HSPC (*n* = 4 healthy cord blood donors) were transduced with LV produced by transient transfection ("transfection", white squares) or by LV-GFP producer cell line ("cell line", black circles) at the indicated MOI and analyzed at 7 (in C) or 14 (in D–F) days after transduction.

G, H Percentage of GFP-positive cells (G) and VCN (H) in T lymphocytes transduced and analyzed as in (A–D) ("cell line", black circles, *n* = 4 transductions using two different LV batches; "transfection", white squares, *n* = 2 transductions).

I, J Human FIX expression (% of normal) in the plasma over time (I) and VCN (J) in liver DNA of hemophilia B mice treated with LV-FIX produced by transient transfection (*n* = 4, white squares) or from producer cell line (*n* = 10 using two different LV batches, black circles). No significant differences.

Data information: In (A–J), data are presented as mean with SEM (for *n* ≥ 3), mean with range (for *n* = 2), and/or single values. Significance was assessed by Mann–Whitney test in (C–F) and (J) or by two-way ANOVA for repeated measures in (I).

---

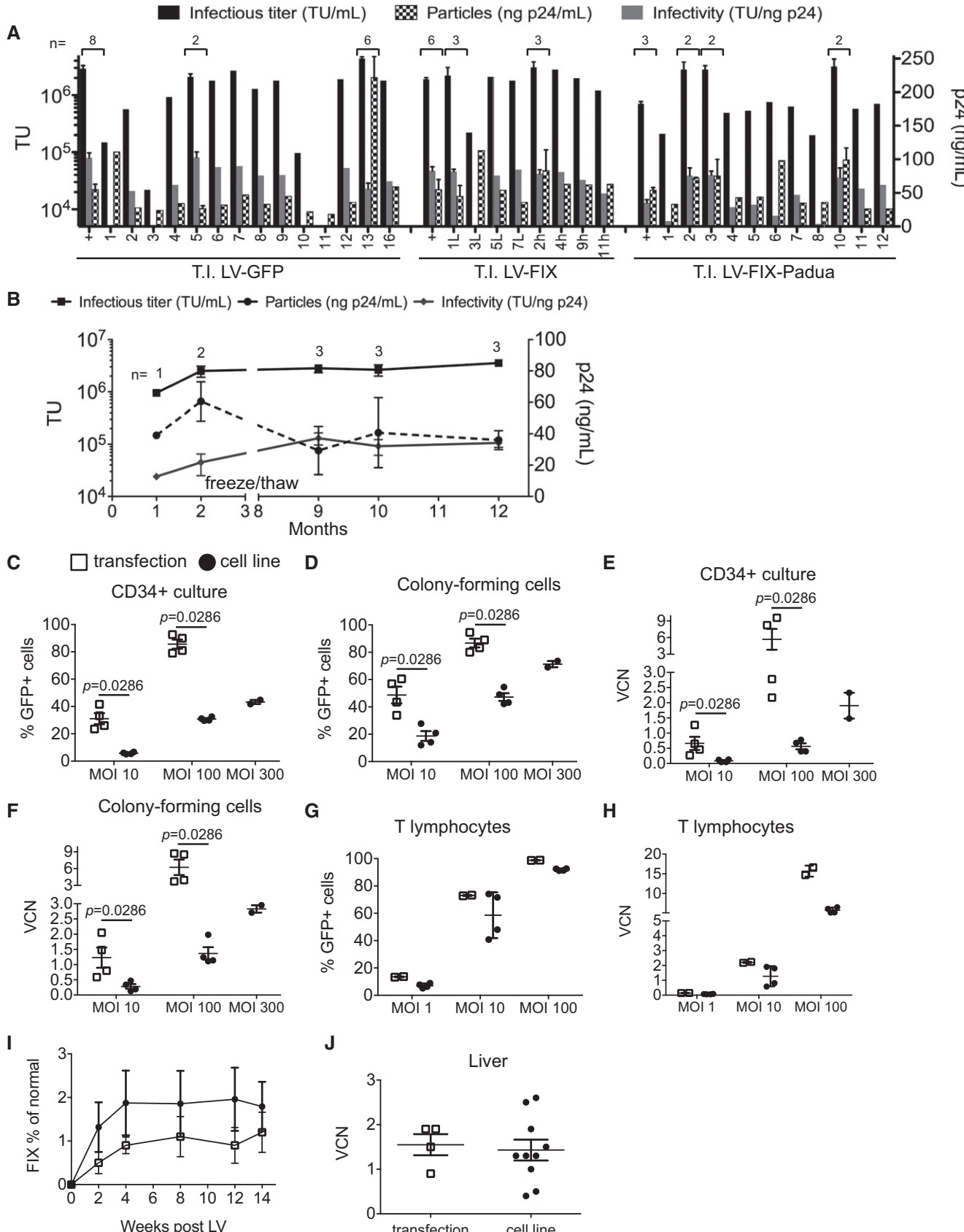

**Figure 2.**

**Table 1.** Purification of cell line-produced LV.

| | Titer | Particles | Infectivity | Volume | Total | |
|---|---|---|---|---|---|---|
| | TU/ml | ng p24/ml | TU/ng p24 | ml | TU | ng p24 |
| Initial | $4.2 \times 10^6$ | 396 | $1.1 \times 10^4$ | 6,000 | $2.5 \times 10^{10}$ | $2.4 \times 10^6$ |
| Final | $1.1 \times 10^8$ | 10,207 | $1.1 \times 10^4$ | 64.5 | $7.2 \times 10^9$ | $6.6 \times 10^5$ |
| Yield | – | – | – | – | 29% | 28% |

The table shows infectious titer, physical particles, and specific infectivity of (i) LV-containing conditioned medium collected from LV-FIX-Padua producer cells 3 days after dox addition (initial material) and (ii) concentrated purified LV formulated in saline solution at 5% dimethyl sulfoxide (final product), according to our previously reported process (Biffi *et al*, 2013; Cantore *et al*, 2015). Note that the yield of the process is in line with results previously reported for LV produced by conventional transient transfection (Aiuti *et al*, 2013; Biffi *et al*, 2013) and that the recovery in specific infectivity is 100%.

ratio among the transduced lymphocytes at any tested dose (Fig EV3D–F). We then intravenously administered $4.5 \times 10^8$ TU/ mouse of LV-FIX derived either from the two most productive cell line clones or from transient transfection to hemophilia B mice and observed stable reconstitution of circulating FIX at 50–100 ng/ml and 1.5 VCN in the liver of treated mice (Fig 2I and J). Importantly, we did not detect any difference in FIX expression or VCN in mice treated with LV produced by either method.

**Improved stability in human serum of stable cell line-produced LV**

We evaluated the stability of LV in heat-inactivated or fresh complement-preserved human serum and observed that LV inactivation became significant only upon dilution below a threshold concentration (Fig 3A and B). We confirmed that LV inactivation was mostly dependent on the heat-labile complement component and subject to donor-to-donor variability (between 15 and 60% recovery of titer at the highest dose tested; Fig 3C), as previously reported (DePolo *et al*, 2000; Schauber-Plewa *et al*, 2005). Complement-mediated LV inactivation was overcome by adding eculizumab, a humanized monoclonal antibody that binds complement protein C5 (Fig 3D; Rother *et al*, 2007; Legendre *et al*, 2013). Interestingly, we found that the cell line-produced LV was 10-fold more resistant to inactivation in human serum (see Fig 3B). Because it has been shown that a

major determinant of LV inactivation is VSV.G, we hypothesized that the increased resistance of cell line-produced LV was due to a reduced content of VSV.G on the envelope of these LV. To test this hypothesis, we produced LV with decreasing amounts of the VSV.G-expressing construct by transient transfection and measured the content of VSV.G on LV particles by immune electron microscopy. The VSV.G content per virion of cell line LV was on average 35% less than that of LV produced by transient transfection with standard amount of VSV.G plasmid (Fig 3E and F). LV with decreasing VSV.G content showed increased resistance to inactivation in human sera and LV produced by transfection with the lowest amount of VSV.G plasmid showed the most similar dose inactivation profile to the cell line-produced LV in this assay (see Fig 3B). We also determined LV inactivation in sera of different species and found that, while mouse sera did not significantly inactivate LV, dog sera showed a slightly stronger inactivation than human serum and that the dose-dependent LV inactivation profile was overlapping for monkey and human sera (Fig 3G), suggesting that monkey models should appropriately predict the human setting, concerning complement-mediated LV inactivation. Overall, these data show that LV inactivation in human serum is dependent on the LV concentration and the amount of VSV.G on the viral particles and it could be potentially rescued by using anti-complement antibody. Moreover, VSV.G-low LV, such as those produced by the stable cell line, are more resistant to complement-mediated inactivation in human

**Figure 3.  Stability of LV in human serum.**

A   LV were incubated for 1 h at 37°C in control medium (no-serum control), complement-preserved or heat-inactivated (1 h at 56°C; H-i) serum, and then titered on 293T cells.

B   Percentage of titer recovered, compared to the no-serum control (20 independent assays performed at the indicated LV concentration) of LV produced by transient transfection with 9 µg/15-cm dish of VSV.G plasmid DNA (black squares, $n = 2$–8 per concentration) or decreasing amounts of VSV.G plasmid DNA (blue to light blue squares, as indicated, $n = 1$ per concentration) or LV produced by LV-GFP or LV-FIX-Padua producer cell line (from bulk-sorted population or most productive clones, green circles, $n = 1$–4 per concentration), incubated with heat-inactivated (empty symbols) or fresh complement-preserved (filled symbols) human sera.

C   Percentage of titer recovered, compared to the no-serum control of matched doses of LV incubated with heat-inactivated (empty squares, $n = 7$ independent assays) or fresh complement-preserved (filled squares) human sera from 12 healthy blood donors (single values, indicated with different colors).

D   Percentage of titer recovered, compared to the no-serum control (at the indicated LV concentration, $n = 3$ for $10^6$, $n = 1$ for $10^5$ TU/ml) of LV incubated with heat-inactivated (H-i, white bars) or fresh complement-preserved human serum without (black bars) or with eculizumab at the indicated concentration (red and pink bars).

E   Representative photomicrographs of LV batches immunostained or not with anti-VSV.G, as indicated and analyzed by electron microscopy. Scale bars: 100 nm.

F   Quantitative analysis (gold particles per virion) of LV produced by transient transfection or by stable producer cell line as in (B) immunostained with anti-VSV.G antibodies or, as staining control, without the primary antibody (ctrl, black triangles), and analyzed by electron microscopy ($n = 45$–91 virions per sample).

G   Percentage of titer recovered, compared to the no-serum control of different concentrations of LV incubated with heat-inactivated (empty symbols) or fresh complement-preserved (filled symbols) human (black squares, same data set as in panel B), monkey (brown squares, $n = 2$–7 per concentration), dog (red squares, $n = 3$–13 per concentration), rat (purple squares, $n = 1$ per concentration), or mouse (yellow squares, $n = 1$–5 per concentration) sera.

Data information: In (B–D, F, G) data are presented as mean with SEM (for $n \geq 3$), mean with range (for $n = 2$), and/or single values. Significance was assessed by Kruskal–Wallis test with Dunn's multiple comparison test in (F).

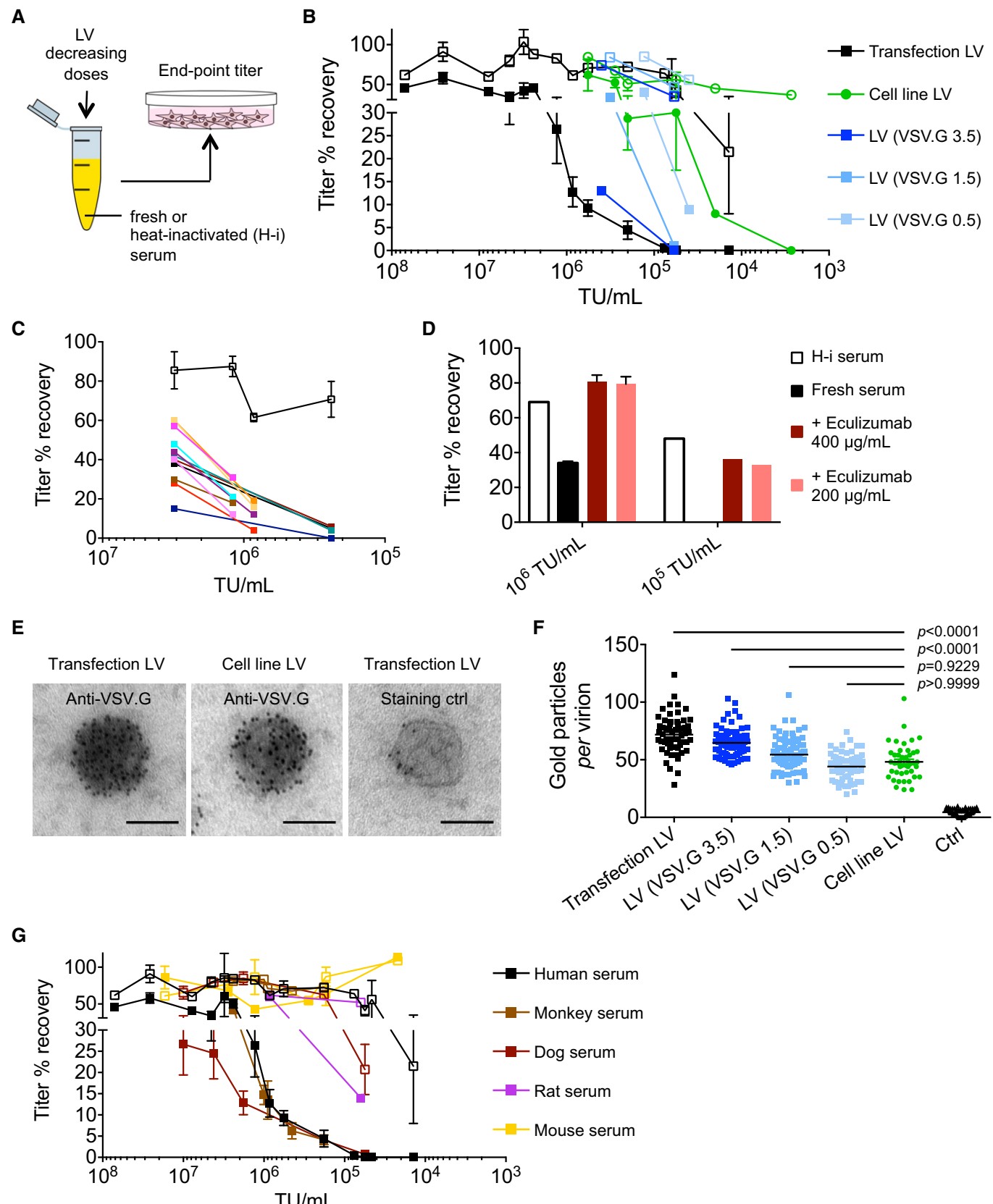

**Figure 3.**

serum. To further investigate whether additional factors are responsible for the higher resistance of the cell line-produced LV to complement-mediated inactivation in human serum, we measured the expression of CD35, CD46, CD55, and CD59 complement regulatory proteins (Garcia *et al*, 2016) in the packaging cell line and in 293T cells used for transient transfection. All these proteins except for CD35 were expressed in both cell types with slightly higher expression of CD46 and CD55 in the former cells (Fig EV4), which might contribute in part to the increased resistance of cell line LV to complement-mediated inactivation.

**Reduced immune activation by MHC-free LV**

MHC-I molecules from the producer cell surface become incorporated in the vector particles, as shown by Western blot of LV lysates and electron microscopy of LV particles immunostained with anti-MHC-I antibodies (Fig 4A–C). Because MHC-I molecules are highly polymorphic and immunogenic across different individuals (Shiina *et al*, 2009), we generated producer cells devoid of surface-exposed MHC-I by permanently disrupting the gene encoding for beta-2 microglobulin (B2M), required for the expression of MHC-I molecules on the cell membrane (Adiko *et al*, 2015). We transiently delivered the Cas9 nuclease (Hsu *et al*, 2014) and single-guide RNA (sgRNA) targeting the first exon or the start codon of the *B2M* gene, to the LV packaging cell line or 293T cells, generally used to produce LV by transient transfection. Up to 44% of the cells lost B2M expression and, as a consequence, MHC-I expression on their membrane (Figs 4D and EV5A–C). These results were confirmed at the genetic level, showing up to 35% of *B2M* alleles bearing indels (Fig EV5B and C). We then enriched for B2M-negative or B2M-positive cells to near purity (> 95%) by FACS. We found no significant differences in infectious titer, particles, and infectivity of LV produced by B2M-negative or B2M-positive sorted cells (Fig EV5D–F). Western blot and electron microscopy on LV produced by B2M-negative cells (MHC-free LV) showed lack of MHC-I antigen (see Fig 4A–C). The absence of MHC

on LV did not affect the level of incorporation of VSV.G (see Fig 4C). FIX output and liver VCN were overlapping in hemophilia B mice treated with LV-FIX-Padua produced in B2M-positive or B2M-negative cells by transient transfection, or by the most productive clone of the B2M-negative producer cell line, further confirming comparable liver gene transfer by LV produced by either cells and methods (Fig 4E and F). MHC-free LV were more resistant to antibody-dependent complement-mediated inactivation than their MHC-bearing counterparts in sera obtained from allo-immunized individuals, such as poly-transfused patients, which contained antibodies against the MHC specificities of 293T cells (Fig 4G). We also observed increased stability of MHC-free LV in human sera when comparing LV pseudotyped with the baculovirus GP64 envelope protein, although these pseudotypes were *per se* more resistant to complement-mediated inactivation than VSV.G pseudotypes, as previously shown (Fig 4H; Schauber *et al*, 2004). We then observed that human primary T cells were significantly less activated when co-cultured with autologous monocytes previously exposed to MHC-free LV than conventional MHC-bearing LV particles, both when testing LV particles pseudotyped with VSV.G or with GP64, although the latter pseudotype is known to show tropism restriction against hematopoietic lineage cells (Fig 4I and J; Schauber *et al*, 2004). These data show that APC exposed to conventional LV present allo-antigens derived from the vector particles that can trigger allogeneic immune responses. Importantly, these responses were substantially decreased by using MHC-free LV particles, independently on the vector pseudotype. Overall, these results show that MHC-I-negative cells, generated by genetic inactivation of *B2M*, produce MHC-free LV that have the same infectivity but lower immunogenicity than conventional LV.

## Discussion

Here, we report the generation of LV with modified surface, achieved by changing the protein composition of the producer cell

---

**Figure 4. Generation, imaging, and evaluation of MHC-free LV.**

A      Western blot on protein extracts from LV batches produced by B2M-positive (LV) or B2M-negative (MHC-free LV) cells.

B      Representative photomicrographs of LV batches immunostained with anti-VSV.G or anti-MHC-I antibodies, as indicated and analyzed by electron microscopy. Scale bars: 100 nm.

C      Quantitative analysis (gold particles per virion) of LV (black squares) or MHC-free LV (orange squares) immunostained with anti-MHC-I antibodies (top panel, *n* = 57–63 virions per sample) or with anti-VSV.G antibodies (bottom panel, *n* = 35–48 virions per sample), or, as staining control, without the primary antibody (ctrl, black triangles), and analyzed by electron microscopy.

D      Flow cytometry analysis (contour plots with outliers) of LV packaging cell line untreated, CRISPR/Cas9 treated, B2M positive or B2M negative sorted as indicated, performed 1 month after sorting.

E, F   Human FIX expression (% of normal) in the plasma (E) over time and VCN (F) in liver DNA of hemophilia B mice treated with LV-FIX-Padua produced by transient transfection into B2M-positive (*n* = 3, black squares or line) or B2M-negative (*n* = 4, orange squares or line) 293T cells, or from B2M-negative producer cells (*n* = 4, orange circles or line). No significant differences.

G, H   Percentage of titer recovered, compared to the no-serum control (5 independent assays performed at the indicated LV concentration) of transient transfection produced VSV.G- (G) or GP64- (H) pseudotyped LV (black lines) or MCH-free LV (orange lines) incubated with heat-inactivated (empty squares) or fresh complement-preserved (filled squares) human sera from allo-immunized individuals, thus containing antibodies against MHC-I (for VSV.G *n* = 1–3 per concentration, for GP64 *n* = 2–5 per concentration).

I, J   Human peripheral blood-derived monocytes were exposed to matched physical particle doses of VSV.G- (J, left panel) or GP64- (J, right panel) pseudotyped LV or MHC-free LV particles, co-cultured with T cells of the same blood donor, and activation of CD3-positive T cells was measured by interferon-γ production (two independent assays with seven healthy blood donors and two different packaging cell line-produced LV or MHC-free LV batches for VSV.G-pseudotyped LV; three independent assays with five healthy blood donors for transient transfection produced GP64-pseudotyped LV or MHC-free LV.

Data information: In (C, E–H), data are presented as mean with SEM (for *n* ≥ 3), mean with range (for *n* = 2), and/or single values. Significance was assessed by Kruskal–Wallis test with Dunn's multiple comparison test in (C, F) or by two-way ANOVA for repeated measures (E) or by Wilcoxon matched pairs test (G, H, J). Source data are available online for this figure.

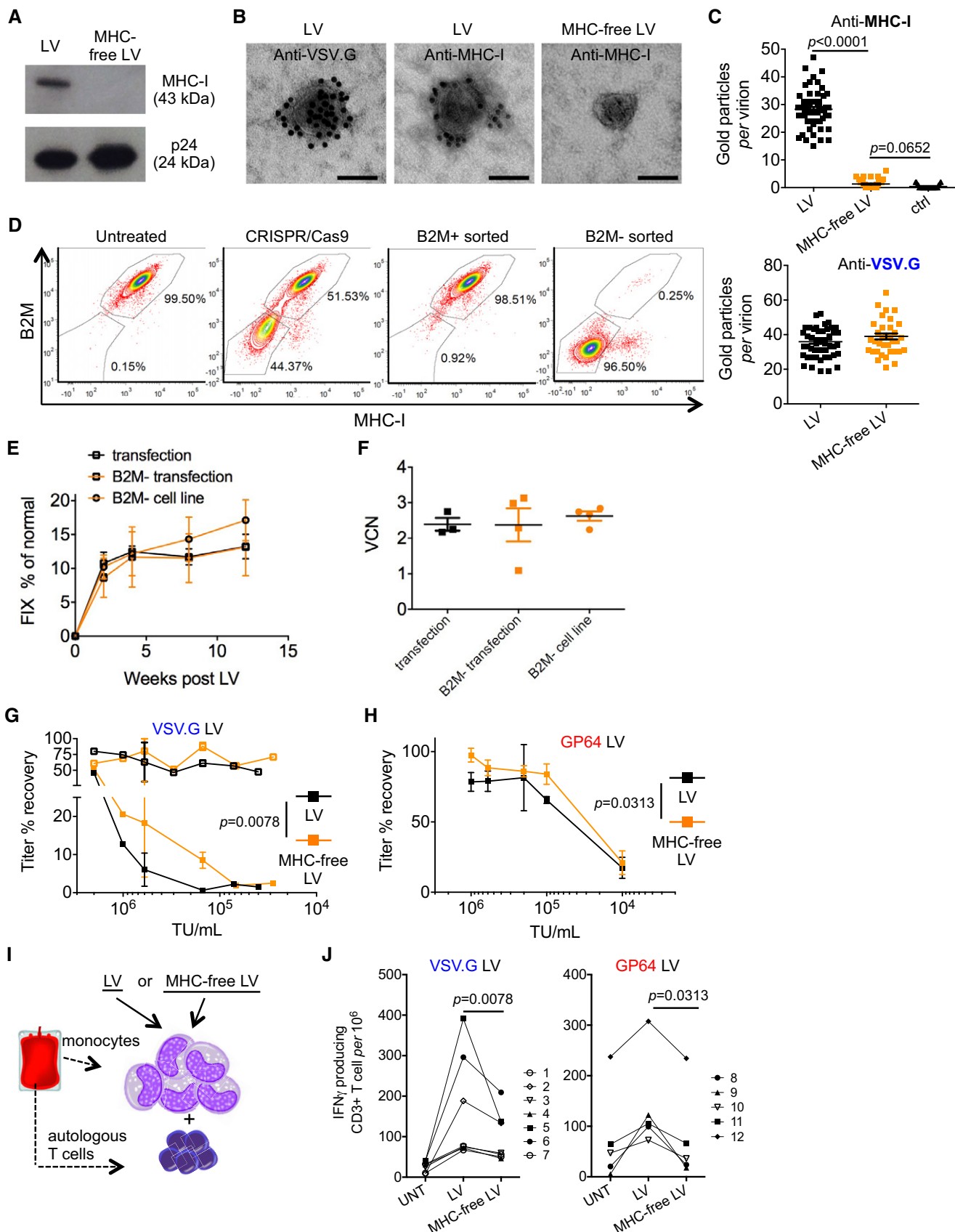

**Figure 4.**

plasma membrane through genome editing. These LV lack polymorphic MHC molecules that may otherwise trigger allogeneic adaptive immune responses. We noted that human primary T cells are activated by autologous monocytes exposed to LV, an effect significantly inhibited by using MHC-free LV, indicating productive presentation of LV-derived allo-MHC antigens. Because MHC molecules of producer cells are incorporated into LV, they might in turn be transferred to the plasma membrane of APC upon LV uptake, or their digested fragments might be presented on class-I and class-II MHC of the APC, thus eliciting an allo-reactive immune response, potentially leading to immune-mediated killing of the target cells shortly after transduction. Furthermore, antibodies against the MHC specificities of the producer cells inactivated LV particles. Such allo-antibodies can be found in patients frequently exposed to blood cell products and might compromise *in vivo* gene transfer and trigger cytotoxicity and phagocytosis of autologous cells shortly after *in vivo* or *ex vivo* transduction by LV. In contrast, MHC-free LV alleviate all these risks. Genetic inactivation of *B2M* in producer cells yielded LV with comparable transduction efficiency to their standard counterparts, in striking contrast to wt HIV, which instead was recently reported to require MHC-I for full-particle infectivity because of its association with the HIV envelope proteins (Serena *et al*, 2017). MHC-free LV are thus better suited for human gene therapy, because of reduced concerns for eliciting immune responses. These results represent a paradigm for further iteration of engineering producer cells to enhance the properties of advanced therapy medicinal products, a task made easy by current genome editing technology.

The improved LV were produced by standard transfection or exploiting novel stable inducible producer cell lines. Several inducible LV packaging cell lines have already been reported, but their reproducible conversion into producer cell lines has been challenging (Kafri *et al*, 1999; Farson *et al*, 2001; Broussau *et al*, 2008; Stewart *et al*, 2009; Throm *et al*, 2009; Stornaiuolo *et al*, 2013). The LV genome has been introduced either by a viral vector, an approach, however unsuitable for SIN vectors, or by stable transfection, which may generate rearrangements and concatemers with low expression and sometimes inhibitory activity on packaging and titers. Here, we show that single-copy integration of a full-length packaging competent LV genome into a validated locus allows consistent generation of high-titer producers, overcoming the need to extensively characterize each new producer cell line for every LV of interest. Although the LV genome is constitutively expressed, its nuclear export and encapsidation are Rev-dependent, thus explaining the lack of detectable infectious LV in the medium of non-induced producers, which protects the cells by LV superinfection. The stability and scalability of these producer cells in culture also reflect the tight regulation of the toxic VSV.G protein. Further development may result in adaptation of these cells to grow in suspension and sustain expansion at industrial scales.

As *ex vivo* T-cell-based gene therapies progress to the market stage, the availability of scalable LV producer cell lines will meet the requirement for larger batch sizes and improved quality and standardization of the therapeutic product. In addition, consistent generation of new LV producers may help addressing the need for rapid development of patient-specific vectors expressing T-cell receptor genes that target tumor neo-antigens. Stable LV producer cell lines may be even more crucial to support the development of *in vivo* gene therapies. We show here that LV produced by our stable cell line are more resistant to complement-mediated inactivation in human sera, due, at least in part, to a reduced content of VSV.G. The classical complement activation pathway is triggered by clustering of antibody–antigen complexes on the target antigen; thus, we speculate that even a relatively small (35%) reduction in the density of VSV.G on the vector envelope can decrease immunoglobulin clustering, resulting in pronounced reduction of complement fixation on the LV particles. While the lower VSV.G content did not impair vector stability upon purification or the efficiency of liver gene transfer, it may explain the lower transduction efficiency of HSPC by the cell line-produced LV, compared to LV produced by transient transfection. LV lacking MHC-I and with low VSV.G content may thus be advantageous for *in vivo* delivery, not only because of the improved resistance to complement-mediated inactivation but also because they are likely to be less opsonized and inflammatory. Altogether, the advances described in this work, by decreasing LV immunogenicity and sensitivity to complement-mediated inactivation and supporting their scalable manufacturing, should improve the feasibility, safety, and efficacy of *in vivo* gene therapy with LV.

## Materials and Methods

### Experimental design

Sample size was chosen according to previous experience with experimental models and assays. No sample or animal was excluded from the analyses. Mice were randomly assigned to each experimental group. Investigators were not blinded.

### Plasmid construction

The linearized HindIII pcDNA5/TO plasmid (Invitrogen) was ligated with HindIII Rev fragment of plasmid pK-Rev (Biffi *et al*, 2013), to obtain pY-Rev. The MluI-NotI restriction fragment of pcDNA5/TO was inserted into MluI-HindIII backbone of plasmid pCMV3.1—neomycin-containing plasmid. The resulting construct was digested with NotI and BamHI and ligated with BamHI-NotI Gag/Pol fragment from pKLGag/Pol (Biffi *et al*, 2013) to obtain pY-Gag/Pol. The VSV.G fragment of plasmid pK.G (Biffi *et al*, 2013) was ligated with backbone of pcDNA5/TO using BamHI and NotI restriction enzymes. The resulting plasmid was double-digested with HindIII-BamHI and subsequently ligated to the HindIII-BamHI puromycin fragment of pSV40-puro (Selexis) to obtain pY-VSV.G. The donor plasmid for the LV-GFP targeting (pLV-GFP) was generated by introducing the AAVS1 left and right homology arms into the NcoI site downstream the SV40 polyA and into the HindIII site upstream the CMV promoter into pCCLsin.PPT.hPGK.GFP.wpre. The donor plasmid for the LV-FIX or LV-FIX-Padua targeting was obtained exchanging the PGK.GFP expression cassette of the LV-GFP donor plasmid with the ET.hFIX.142T expression cassette or ET.co-hFIX-Padua of pCCLsin.PPT.ET.hFIX.wpre.142-3pT or pCCLsin.PPT.ET.co-hFIX-Padua.Wpre.142-3pT plasmid (HpaI-KpnI), respectively (Cantore *et al*, 2015). The Cas9 and sgRNA-expressing plasmids were previously described (Amabile *et al*, 2016). The sequences of

the CRISPR used to generate the sgRNA are as follows: *B2M* exon 1 (GAGTAGCGCGAGCACAGCTAAGG), *B2M* start codon (GGCCACGG AGCGAGACATCTCGG).

## Packaging cell line generation

At Baxter, 293 T-REx cells (Invitrogen) were cultivated in Dulbecco's modified Eagle's medium (DMEM, Invitrogen) with 10% fetal calf serum (Invitrogen), 2 mM glutamine (Invitrogen), and 5 mg/l blasticidin S (Invitrogen); 293 T-REx cells were transfected by electroporation using 5 μg of pY-Rev, following manufacturer's instructions (PEQLAB, Microporator MP-100), selected in the presence of 5 mg/l blasticidin S and 50 mg/l hygromycin B (Invivogen); 293 T-REx Rev cells were then transfected as above with pY-Gag/Pol plasmid, selected in the presence of 5 mg/l blasticidin S, 50 mg/l hygromycin B, 2.5 mg/l Geneticin (Invitrogen) and were then transfected with pY-VSV.G plasmid as above and selected in the presence of 5 mg/l blasticidin S, 50 mg/l hygromycin B, 2.5 mg/l geneticin, 5 mg/l puromycin (Sigma) to generate the packaging cell line.

## Site-specific integration and gene disruption

Targeted integration in *AAVS1* was performed by calcium phosphate-mediated transient transfection of the indicated amount of the desired donor plasmid and the ZFN-expressing plasmid (Lombardo *et al*, 2011). Gene disruption was performed by calcium phosphate-mediated transient transfection of the indicated amount of the desired sgRNA-expressing plasmid and the Cas9-expressing plasmid.

## LV production

VSV.G-pseudotyped third-generation self-inactivating (SIN) LV were produced by calcium phosphate transient transfection into 293T cells, or by LV packaging or producer cell lines. 293T cells were transfected with a solution containing a mix of the selected LV genome transfer plasmid, the packaging plasmids pMDLg/pRRE and pCMV.REV, pMD2.G or pBA-AcMNPV-gp64 (Schauber *et al*, 2004) and pAdVantage (Promega), as previously described (Cantore *et al*, 2015). Medium was changed 14–16 h after transfection and supernatant was collected 30 h after medium change. Alternatively, LV production was induced when LV producer or packaging cells were in a sub-confluent state, by replacing the culture medium with medium containing doxycycline (Sigma) 1 μg/ml and supernatant was collected 3 days after induction. LV-containing supernatants were passed through a 0.22-μm filter (Millipore) and, when needed, transferred into sterile poliallomer tubes (Beckman) and centrifuged at 20,000 *g* for 120 min at 20°C (Beckman Optima XL-100K Ultracentrifuge). LV pellet was dissolved in the appropriate volume of PBS to allow 500–1,000× concentration. LV purification from large-scale (6,000 ml) production was performed as described (Biffi *et al*, 2013; Cantore *et al*, 2015).

## LV titration

For LV titration, 100,000 293T cells were transduced with serial LV dilutions in the presence of polybrene (8 μg/ml). For LV-GFP, cells were analyzed by flow cytometry 3–7 days after transduction and infectious titer, expressed as transducing units$_{293T}$ (TU)/ml, was calculated using the formula TU/ml = ((% GFP$^+$ cells/100) *100,000*(1/dilution factor)). For all other LV, genomic DNA (gDNA) was extracted 14 days after transduction, using Maxwell 16 Cell DNA Purification Kit (Promega), following manufacturer's instructions. VCN was determined by quantitative PCR (qPCR) starting from 100 ng of template gDNA using primers (HIV fw: 5′-T ACTGACGCTCTCGCACC-3′; HIV rv: 5′-TCTCGACGCAGGACTCG-3′) and a probe (FAM 5′-ATCTCTCTCCTTCTAGCCTC-3′) against the primer binding site region of LV. The amount of endogenous DNA was quantified by a primers/probe set against the human telomerase gene (Telo fw: 5′-GGCACACGTGGCTTTTCG-3′; Telo rv: 5′-GGTGAACCTCGTAAGTTTATGCAA-3′; Telo probe: VIC 5′-TCAGG ACGTCGAGTGGACACGGTG-3′ TAMRA). VCN was calculated by the formula = (ng LV/ng endogenous DNA)* number of LV integrations in the standard curve. The standard curve was generated, by using a CEM cell line stably carrying one vector integrant, which was previously determined by Southern blot and fluorescent *in situ* hybridization (FISH). All reactions were carried out in duplicate or triplicate in a Viia7 Real Time PCR thermal cycler (Applied Biosystems). Each qPCR run carried an internal control generated by using a CEM cell line stably carrying four vector integrants, which were previously determined by Southern blot and FISH analysis. Infectious titer, expressed as TU/ml, was calculated using the formula TU/ml = (VCN*100,000*(1/dilution factor)). LV physical particles were measured by HIV-1 Gag p24 antigen immunocapture assay (Perkin Elmer) following manufacturer's instructions. LV specific infectivity was calculated as the ratio between infectious titer and physical particles.

## Cell cultures

293T, LV packaging and producer cell lines were maintained in Iscove's modified Dulbecco's medium (IMDM, Sigma) supplemented with 10% fetal bovine serum (FBS, Euroclone), 4 mM glutamine (Lonza), penicillin, and streptomycin 100 IU/ml (Lonza). Human cord blood-derived CD34-positive cells were purchased from Lonza and kept in culture at the concentration of $10^6$ cells/ml in Stem Spam SFEM (Stem Cell Technologies) supplemented with human cytokines purchased from PeproTech, thrombopoietin (TPO, 100 ng/ml), interleukin-6 (IL-6, 100 ng/ml), stem cell factor (SCF, 20 ng/ml), Flt3 ligand (FLT3L, 20 ng/ml) for at least 7 days, then cultured in IMDM supplemented with 10% FBS, 4 mM glutamine, penicillin, and streptomycin 100 IU/ml to prompt differentiation. The CFC assay was performed as described (Escobar *et al*, 2014). Human cord blood mononuclear cells enriched for CD3-positive T cells were kept in culture at a concentration of $10^6$ cells/ml in IMDM supplemented with interleukin 7 (5 ng/ml), interleukin 15 (5 ng/ml), 10% FBS, 4 mM glutamine, penicillin, and streptomycin 100 IU/ml. T lymphocytes were pre-stimulated *in vitro* with beads coated with anti-human CD3 and anti-human CD28 (3 beads/cell, Dynabeads human T-activator CD3/CD28, Invitrogen). Four days after transduction, beads were mechanically detached and removed using MagnaRack (Invitrogen). Primary human monocytes and lymphocytes were magnetically sorted (Miltenyi Biotec) from buffy coats of healthy donors. All cells were maintained in a 5% $CO_2$ humidified

atmosphere at 37°C. All cell lines were routinely tested for myco-plasma contamination.

### AAVS1-LV junction detection

The presence of junctions between the AAVS1 genomic locus and the 3′ and 5′ of the donor DNA construct was tested in gDNA extracted using Maxwell 16 Cell DNA Purification Kit (Promega) following manufacturer's instructions. For the LV-GFP targeting in the AAVS1 site, we set a PCR (AmpliTaq Gold DNA polymerase system by Applied Biosystems) using primers for detecting the 5′ junction (AAVS1 fw: 5′-AACTCTGCCCTCTAACGCTGC-3′; ΔNef rv: 5′-CGAGCTCGGTACCTTTAAGACC-3′) and for detecting the 3′ junction (Gag fw: 5′-GAGTCCTGCGTCGAGAGAG-3′; AAVS1 rv: 5′-AAC GGGGATGCAGGGGAACG-3′). For LV-FIX or LV-FIX-Padua targeting in the AAVS1 site, we set a PCR using primers for detecting the 5′ junction (AAVS1 fw: 5′-AACTCTGCCCTCTAACGCTGC-3′; GFP rv: 5′-GTCTTGTAGTTGCCGTCGTCC-3′) and for detecting the 3′ junction (Gag fw: 5′-GAGTCCTGCGTCGAGAGAG-3′; AAVS1 rv: 5′-AA CGGGGATGCAGGGGAACG-3′). PCR products were analyzed on a 1% agarose gel.

### Mismatch-selective endonuclease assay

The mismatch-selective endonuclease assay was used to measure the extent of mutations consequent to non-homologous end joining (NHEJ) at the Cas9 target sites, as described (Lombardo *et al*, 2011). PCR was performed using primers flanking the sgRNA binding site in the *B2M* gene (fw: 5′-TACAGACAGCAAACTCACC CAGTC-3′; rv: 5′-AGAACTTGGAGAAGGGAAGTCACG-3′). The PCR product was denatured, allowed to re-anneal, and digested with Surveyor nuclease assay (Transgenomic). Because this enzyme cuts DNA at sites of duplex distortions, the products of re-annealing between wild-type and mutant alleles (carrying mutations or deletions consequent to the nuclease activity) are specifically digested. The reaction products were separated on a Spreadex EL1200 Wide Mini gel (Elchrom Scientific), stained by ethidium bromide or GelRed (Biotium), and the intensity of the bands was quantified by ImageQuant TL 5 software. The ratio of the uncleaved parental fragment to the two lower migrating cleaved products was calculated using the formula $(1-(\text{parental fraction})^{1/2})*100$.

### DNA copy determination in LV packaging and producer cells

*Rev*, *Gag*, *VSV.G* DNA copies were determined by ddPCR on gDNA extracted using Maxwell 16 Cell DNA Purification Kit (Promega), starting from 5 to 20 ng of template gDNA and using primers/probe sets against the desired cDNA (Rev fw: 5′-CCTTAGCACTTATCTGG GACG-3′; Rev rv: 5′-TAAGTCTCTCAAGCGGTGGT-3′; Rev probe: FAM 5′-TGCGGAGCCTGTGCCTCTTCAGC-3′; Gag fw: 5′-ATCAAG CAGCCATGCAAATG-3′; Gag rv: 5′-CCTTGGTTCTCTCATCTGGCC-3′; Gag probe: FAM 5′-TGCATCCAGTGCATGCAGGGCC-3′; VSV fw: 5′-GCTTCCCTCCTCAAAGTTGT-3′; VSV rv: 5′-GGAGTCACCTGGA CAATCAC-3′; VSV probe: FAM 5′-TGCAACTGTGACGGATGCC GAAGCA-3′). The amount of endogenous DNA was quantified by a primers/probe set against the human telomerase gene, as above. The PCR was performed with each primer (900 nM) and the probe

(250 nM, 500 nM for Rev) following manufacturer's instructions (Bio-Rad), read with QX200 reader, and analyzed with QuantaSoft software (Bio-Rad). The LV genome copies were determined by qPCR as described above (see "LV titration"). To detect possible ZFN integrants, qPCR (Applied Biosystems) was performed with a primers/probe set against the FOK domain of ZFN (FOK fw: 5′-CC TGACGGCGCCATCTAT-3′; FOK rv: 5′-CGATCACGCCGTAATCGAT-3′; FOK probe: FAM 5′-CAGTGGGCAGCCC-3′). A standard curve of plasmid containing the FOK domain was diluted into gDNA of untransduced 293T cells. FOK copies in samples were quantified on this standard curve. gDNA content was determined by using a standard curve obtained with serial dilutions of gDNA extracted from a human cell line. The amount of endogenous DNA was quantified by a primers/probe set against the human telomerase gene, as above. Since 200 ng of gDNA corresponds to 60,000 genomes, it was possible to calculate the number of FOK copies/diploid genomes.

### Human cell experiments

HSPC and T lymphocytes transduction was performed at the indicated MOI (TU/cell). Purified CD14-positive monocytes (purity > 95%) were exposed to MHC-free or control MHC-bearing LV particles produced without vector genome (1 μg of HIV Gag p24/$10^6$ CD14-positive cells) for 16 h for VSV.G-pseudotyped LV, or spinoculated for 2 h at 1,100 *g* at 37°C, or left untreated as control. LV-exposed monocytes were washed three times and then co-cultured for 48 h with autologous purified T cells (purity > 95%) at 1:1 ratio in X-vivo-15 (Lonza) supplemented with 5% human serum, penicillin, and streptomycin 100 IU/ml. T cells secreting IFN-γ in response to autologous LV-exposed monocytes were enumerated by conventional elispot assay using purified anti-human IFN-γ (clone1-D1K, Mabtech) as capture and biotin-conjugated anti-human IFN-γ (clone7-B6-1, Mabtech) as detection antibodies. Spots were counted by ELI.Expert Elispot reader and analyzed by Eli.Analyse sofware (A.EL.VIS). When comparing MHC-free or MHC-bearing LV in this assay, we produced both LV in parallel from B2M-MHC double-positive or double-negative cells sorted from the same population treated for *B2M* disruption (stable packaging cell line or 293T for use in transient transfection) in order to minimize variations not directly related to MHC content.

### Flow cytometry

Flow cytometry analyses were performed using a FACSCanto analyzer (BD Biosciences), equipped with DIVA Software. Between 100,000 and 500,000 cells were harvested, washed with PBS or MACS buffer (PBS pH 7.2 0.5% BSA, 2 mM EDTA), treated with Fc Receptor-Block (Miltenyi Biotec) when antibody stained, and then resuspended in the buffer used for washing. Staining was performed in MACS buffer, incubating cells with antibodies (in the proportion indicated in the table below) for 20 minutes at 4°C in the dark. For vitality staining, 7-aminoactino-mycin D (7AAD, Sigma) was used. Anti-murine IgG beads were used for single-staining controls (BD Biosciences). Rainbow beads (BD Biosciences) were used to calibrate the instrument detectors, for consistent MFI measurement, for analysis performed at different times.

| Antigen | Fluorochrome | Clone | Company | Dilution |
|---------|--------------|-------|---------|----------|
| CD33 | BV421 | WM53 | BD Biosciences | 1:20 |
| CD235a | APC | REA175 | Miltenyi Biotec | 1:20 |
| CD34 | VioBlue | AC136 | Miltenyi Biotec | 1:25 |
| CD3 | PE-Cy7 | HIT3a | BioLegend | 1:50 |
| CD4 | Pacific Blue | RPA-T4 | BioLegend | 1:50 |
| CD8 | APC-Cy7 | SK1 | BD Biosciences | 1:33 |
| CD35 | PE | E11 | BD Biosciences | 1:50 |
| CD46 | PE | 8E2 | eBioscience | 1:20 |
| CD55 | PE | JS11 | BioLegend | 1:33 |
| CD59 | PE | p282 (H19) | BioLegend | 1:33 |
| B2M | PE | 2M2 | BioLegend | 1:20 |
| MHC-I | APC | W6/32 | Santa Cruz Biotech | 1:10 |

## Mice experiments

Founder C57BL/6 *F9* knockout mice were originally obtained from the laboratory of Dr. Inder Verma at the Salk Institute (Wang *et al*, 1997). All the mice were maintained in specific pathogen-free conditions. Vector administration was carried out in adult (7- to 10-week-old) male or female mice by tail vein injections. When both male and female were used in the same experiment, they were equally distributed between different treatment groups. Mice were bled from the retro-orbital plexus using capillary tubes, and blood was collected into 0.38% sodium citrate buffer, pH 7.4. Mice were anesthetized with tribromoethanol (Avertin) and euthanized by $CO_2$ inhalation at the expected time points. All animal procedures were performed according to protocols approved by the Institutional Animal Care and Use Committee.

## Enzyme-linked immunosorbent assay (ELISA)

The concentration of human FIX was determined in mouse plasma by ELISA specific for human FIX antigen (Asserachrom IX:Ag, Stago) following manufacturer's instructions. Absorbance of each sample was determined at a spectrophotometer, using a Multiskan GO microplate reader (Thermo Fisher Scientific) and normalized to antigen standard curves.

## LV inactivation assay

Human serum samples were obtained from 12 healthy donors from San Raffaele Hospital or purchased (Sigma). To test MHC-free LV, human serum samples were obtained from four allo-immunized patients from San Raffaele Hospital, upon obtaining informed consent, according to the principles set out in the WMA Declaration of Helsinki and the Department of Health and Human Services Belmont Report. Sera were thawed, and half of each serum sample was heated at 56°C for 1 h to inactivate the complement. LV were diluted in Iscove's modified Dulbecco's medium (IMDM, Sigma) supplemented with 10% fetal bovine serum (FBS, Euroclone). Twenty microliters of LV was diluted 1:5 into fresh or heat-inactivated serum (or IMDM 10% FBS as the no-serum

control), and the mixture was incubated at 37°C for 1 h. Following incubation, medium was added to the reaction and then serially diluted and used to transduce 293T cells for end-point infectious titer determination, as described (Cantore *et al*, 2015). The titer value was divided by the titer determined for the LV mixed with medium (the no-serum control) and reported as the percentage of recovery of titer compared to this control. When comparing MHC-free or MHC-bearing LV in this assay, we produced both LV in parallel by transient transfection in B2M-MHC double-positive or double-negative cells sorted from the same population treated for *B2M* disruption in order to minimize variations not directly related to MHC content. Eculizumab (commercial name Soliris, produced by Alexion Pharmaceuticals) was obtained from leftovers of patients' doses at the San Raffaele Hospital and added to the LV:serum mix at the indicated concentration. Mouse serum was purchased (Sigma), dog serum was kindly provided by Dr. Timothy Nichols (University of North Carolina), and monkey serum was kindly provided by Dr. Eduard Ayuso (University of Nantes).

## Electron microscopy

Few microliters of concentrated LV batches were absorbed on glow-discharged carbon-coated formvar copper grids and fixed for 20 min with 8% paraformaldehyde in PBS. After several washes in 50 mM glycine in PBS, grids was blocked in 1% BSA in PBS and incubated with primary antibodies diluted in blocking buffer for 30–90 min (Anti-VSV.G, KeraFAST, 1:50, Anti-MHC-I, Santa Cruz Biotech, 1:20). After several washes in 0.1% BSA in PBS, samples were incubated for 30 min with Protein A-gold (5 or 10 nm), fixed with 1% glutaraldehyde, stained with 2% uranyl acetate, or a mix of 0.4% uranyl acetate and 1.8% methyl-cellulose, then air-dried. Grids were observed with a Zeiss LEO 512 transmission electron microscope. Images were acquired by a 2 k × 2 k bottom-mounted slow-scan Proscan camera controlled by EsivisionPro 3.2 software. For quantification of labeling density, random images of viral particles were taken at nominal magnification of 16k and gold particles associated to virions were manually counted using ImageJ. Virions were defined based on expected size (approximately 120 nm) and an electron-dense core.

## Western blot

Total proteins in LV batches were extracted with membrane-protein lysis buffer (150 mM Tris–HCl, 150 mM NaCl, EDTA 5 mM, 1% deoxycholate, 0.1% SDS, 1% Triton X-100) supplemented with PIC (Protease Inhibitor Cocktail, Roche). Samples were resuspended in the lysis solution and incubated at 4°C for 10 min. Lysates were assayed for protein concentration using Bradford assay (Bio-Rad). Twenty micrograms of proteins was run on SDS–PAGE under reducing conditions. For immunoblotting, proteins were transferred to polyvinylidene difluoride (PVDF) membranes using iBlot Gel Transfer stacks (Novex), incubated with the specific antibody followed by peroxidase-conjugated secondary antibodies (ECL Mouse or Rabbit IgG GE Healthcare), and detected using chemiluminescent reagents (ECL, GE Healthcare) and exposure to autoradiography films. The following antibodies were used: rabbit monoclonal anti-human MHC-I (OriGene Technologies, 1:1,000 in TBS, Tween-20 0.1%, skim milk powder 5%) and mouse monoclonal anti-Gag p24 (NIH

AIDS reagent program #3537, 1:1,000 in TBS, Tween-20 0.1%, skim milk powder 5%).

Lentiviral vectors packaging cells ($10^5$) were lysed for 5 min at 95°C in NuPAGE®LDS sample buffer (Thermo Fisher Scientific) and run on SDS–PAGE and then blotted on nitrocellulose membrane (Thermo Fisher Scientific). After blocking the membrane for half an hour in 5% dry milk powder, Rev and VSV.G proteins were stained by the primary antibodies against HIV-1 Rev 1:200 (Santa Cruz Biotechnology) and anti-VSV.G 1:1,000 (Abcam), respectively, and the secondary HRP-labeled anti-mouse IgG1 antibody 1:50,000 (GE Healthcare), all diluted in TBS Tween-20 0.1%. As internal control, the beta-actin housekeeping gene was stained by anti-beta-actin antibodies 1:3,300 (Novus Biologicals) and HRP-labeled anti-rabbit IgG1 1:50,000 (Thermo Fisher Scientific).

### Vector Copy Number (VCN) determination in transduced cells

For human HSPC experiments, DNA was extracted from both liquid culture and colony-forming cell (CFC) assay, using DNeasy Blood & Tissue Kit (Qiagen) following manufacturer's instructions. For human T-cell experiments, gDNA was extracted using Maxwell 16 Cell DNA Purification Kit (Promega). For mice experiments, DNA was extracted from whole liver samples using Maxwell 16 Tissue DNA Purification Kit (Promega). Vector Copy Number was determined in HSPC and T cells as previously described (Escobar *et al*, 2014). Vector Copy Number in murine DNA was determined by droplet digital (dd)PCR, starting from 5 to 20 ng of template gDNA using primers (HIV fw: 5′-TACTGACGCTCTCGCACC-3′; HIV rv: 5′-TCTCGACGCAGGACTCG-3′) and a probe (FAM 5′-ATCTCTCTCC TTCTAGCCTC-3′) against the primer binding site region of LV. The amount of endogenous murine DNA was quantified by a primers/ probe set against the murine *sema3a* gene (Sema3A fw: 5′-ACC GATTCCAGATGATTGGC-3′; Sema3A rv: 5′-TCCATATTAATGCAG TGCTTGC-3′; Sema3A probe: HEX 5′-AGAGGCCTGTCCTGCAGCTC ATGG-3′ BHQ1). The PCR was performed with each primer (900 nM) and the probe (250 nM) following manufacturer's instructions (Bio-Rad), read with QX200 reader and analyzed with Quanta-Soft software (Bio-Rad).

### Statistical analysis

Statistical analyses were performed upon consulting with professional statisticians. When normality assumptions were not met, nonparametric statistical tests were performed. Mann–Whitney or Kruskal–Wallis tests were performed when comparing two or more experimental groups, respectively. For paired observations, Wilcoxon matched pairs test was performed.

**Expanded View** for this article is available online.

## Acknowledgements

We thank S. Marktel (San Raffaele Hematology and Bone Marrow Transplantation Unit) for serum samples from allo-immunized patients; T. Plati and A. Migliara for help with some experiments; MolMed S.p.A. for large-scale production using the LV producer cell line; A. Nonis, A. Pramov, and C. Di Serio for statistical consulting. T. Liu and R. Peters (Bioverativ) and all other members of the Naldini, Lombardo, and Gentner laboratories for helpful discussions. We thank the ALEMBIC facility at the San Raffaele Scientific

---

**The paper explained**

**Problem**

Lentiviral vectors (LV) are attractive vehicles for gene therapy. However, their *in vivo* administration remains particularly challenging, as it requires high quantity and quality of vector and raises concerns for the activation of innate and adaptive immune responses, which may be detrimental for both the safety and efficacy of the therapy. Moreover, carryover of allogeneic histocompatibility complexes on LV particles from the human vector producer cells may contribute to these immune responses.

**Results**

Here, we generate stable cell lines, which support scalable and consistent production of LV that are capable of efficient liver gene transfer and are more resistant to complement-mediated inactivation in human sera. Moreover, by genetically inactivating the beta-2 microglobulin gene in LV producer cells, we obtain LV lacking class-I major histocompatibility complex on their surface, which maintain gene transfer capacity and escape immune recognition by human T cells.

**Impact**

The advances described in this work, by supporting scalable manufacturing of LV and decreasing their immunogenicity and sensitivity to complement-mediated inactivation, should facilitate their application to *in vivo* gene therapy for hemophilia and other diseases. Moreover, we show that targeted genome editing of producer cells can be applied to improve the properties of gene therapy vectors, an approach which may have broad application in molecular medicine.

Institute for help with electron microscopy analysis. M.M. conducted this study as partial fulfillment of her International Ph.D. Course in Molecular Medicine at San Raffaele University, Milan. This work was supported by Telethon (SR-Tiget Core Grant 2011-2016), and Bioverativ sponsored research agreement.

## Author contributions

MM designed and performed experiments, analyzed and interpreted data, and contributed to writing the manuscript. AA designed and performed experiments and interpreted data concerning T-cell responses. SB, MB, FR, and TDT performed experiments and analyzed data. AR performed electron microscopy experiments. JL performed experiments and analyzed data concerning the original LV packaging cell-line generation. FS supervised JL research. MCH provided reagents. AL provided intellectual input and reagents, interpreted data on genome editing experiments, and edited the manuscript. LN and AC designed and supervised research, interpreted data, and wrote the manuscript. LN provided overall coordination and financial support.

## Conflict of interest

L.N. is inventor on patents on LV technology and microRNA-regulated LV (gene vector, WO2007000668). A.C., A.L., and L.N. are inventors on a filed patent on MHC-negative LV producer cells. These patents are owned by Telethon Foundation and San Raffaele Scientific Institute. The LV and reagents described in this manuscript are available to interested scientists upon signing a MTA with standard provisions. There are some restrictions concerning research involving LV for the gene therapy of hemophilia, except for research aimed at reproducing the findings reported in this manuscript, according to a collaboration agreement between Telethon Foundation, the San Raffaele Scientific Institute, and Bioverativ.

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
