## [Review Process File · EMBO Molecular Medicine]

Genome Editing for Scalable Production of Alloantigen-free Lentiviral Vectors for *in Vivo* Gene Therapy

Michela Milani, Andrea Annoni, Sara Bartolaccini, Mauro Biffi, Fabio Russo, Tiziano Di Tomaso, Andrea Raimondi, Johannes Lengler, Michael C. Holmes, Friedrich Scheifflinger, Angelo Lombardo, Alessio Cantore, Luigi Naldini

Corresponding author: Luigi Naldini, San Raffaele Telethon Institute for Gene Therapy

Review timeline:

Submission date:	30 June 2017
Editorial Decision:	14 July 2017
Revision received:	21 July 2017
Accepted:	26 July 2017

Transaction Report:

(Note: This manuscript was transferred from another journal where it was originally reviewed. Since the original reviews are not subject to EMBO's transparent review process policy, the reports and author response cannot be published.)

Editor: Céline Carret

1st Editorial Decision

14 July 2017

Thank you for the submission of your revised manuscript to EMBO Molecular Medicine. As discussed, and based on the initial set of reviews you provided us with, we have evaluated your revised article and asked an expert external advisor to look at your paper and responses to the referees (comments pasted below). We have now received our advisor's comments and following editorial discussions, including with our chief editor, we have decided to accept your manuscript pending the following final editorial amendments:

1) Animal work and ethical statements:

-please provide the gender of mice used

-human samples: please indicate where you got them from when not purchased (hospital?) and confirm in the main text compliance to the principles set out in the WMA Declaration of Helsinki and the Department of Health and Human Services Belmont Report. .

2) Source Data:

We now encourage the publication of source data, particularly for electrophoretic gels, blots, but also microscopy images with the aim of making primary data more accessible and transparent to the reader. Would you be willing to provide a PDF file per figure that contains the original, uncropped and unprocessed scans of all or key gels used in the figure? The PDF files should be labeled with the appropriate figure/panel number (1 file/figure), and should have molecular weight markers; further annotation may be useful but is not essential. The PDF files will be published online with the article as supplementary "Source Data" files. If you have any questions regarding this just contact me.

***** Advisor's comments *****

I read carefully the paper by Milani et al. It is a very good paper, carefully written and reporting solid and beautifully controlled data. There is very little I would suggest to improve the paper. Frankly, I don't see the point of asking for in vivo data, they would be anyway irrelevant in proving the immunogenicity and complement sensitivity of the viral particles produced by these stable cell lines in a human situation. The only real problem with these lines is their endpoint titer, which is one log lower than that achieved by transient transfection in a typical GMP manufacturing process. For in vivo applications, this generates serious manufacturing problems, since there would be the need to produce ten-fold higher volumes and concentrate the supernatant ten times more compared to a transient transfection system, a formidable challenge in terms of process scalability and cost.

1st Revision - authors' response

21 July 2017

We have now submitted a revised version of the manuscript, according to all your requests below.

We have also updated figure EV4 with improved flow cytometry analysis of LV packaging cell line and adjusted the corresponding sentence in the text.

Corresponding Author Name: Luigi Naldini
Journal Submitted to: EMBO Molecular Medicine
Manuscript Number: EMM-2017-08148P